# Characteristics of spatiotemporal distribution of HIV-1 Gag-containing complexes on the dorsal membrane tracking with live confocal imaging

Xiao Wang[1]☯, Xinye An[1]☯, Hongmei Zhao[1], Dakang Sun [2]*

**1** Laboratory of Clinical Medicine, Binzhou Medical University Hospital, Binzhou, Shandong Province, China, **2** Medical Research Center, Binzhou Medical University Hospital, Binzhou, Shandong Province, China

☯ These authors contributed equally to this work.
* sdkaaa@163.com

## Abstract

### Introduction

In recent years, novel detection methods have provided significant insights into the real-time morphological changes occurring during the HIV life cycle within host cells. However, the detailed dynamics of virus assembly and release, particularly from the perspective of the dorsal cell membrane, remain poorly understood.

### Methods

HEK293T cells were transfected with pEGFP-N3-Gag plasmids, and the spatiotemporal distribution of Gag-EGFP was monitored using the "xyt" or "xyz" imaging modes of laser confocal microscopy (LCM). The motion trajectory of GCC in living cells was manually tracked using the MtrackJ plugin in ImageJ 1.54p. 3D reconstruction of target proteins were processed with the Volume Viewer plugin in ImageJ 1.54p.

### Results

The results revealed that Gag-EGFP proteins exhibited directional movement toward the plasma membrane, where they assembled into Gag-containing complexes (GCCs) of varying sizes and displayed localized small-scale displacements. Furthermore, GCCs on the cell membrane were observed to detach from the dorsal membrane and were subsequently released into the extracellular environment within 3–8 minutes. Three-dimensional (3D) reconstruction demonstrated that Gag-EGFP proteins exist in three distinct forms: granule-like structures, cord-like structures, and giant polymers on the dorsal membrane or in the cytoplasm.

**Data availability statement:** All relevant data are within the paper and its Supporting Information files.

**Funding:** This work was supported by Projects of medical and health technology development program in Shandong province (grant numbers 2018WS545).

**Competing interests:** The authors declare no competing interests.

## Conclusion

Overexpressed Gag proteins on the dorsal cell membrane can form dynamic polymers of various sizes that undergo localized movement or detach from the membrane at different rates. The small spots of GCCs observed in a single confocal plane may actually have significantly larger sizes and unexpected shapes within the cell.

## Introduction

HIV virus infection can damage the body's immune system, leading to an extremely serious health issue. In 2019, there were approximately 38 million HIV infected individuals worldwide, and in 2022, approximately 630000 people died from HIV related illnesses [1]. The Gag capsid precursor protein, also known as Pr55 Gag, plays a pivotal role in viral assembly and release. This polyprotein is responsible for forming the structural core of the virus, and its proper function is essential for generating infectious viral particles. The Gag protein comprises four structural domains: matrix (MA), capsid (CA), nucleocapsid (NC), and p6 [2,3]. These domains can be cleaved by an activated protease [4]. The N-terminal matrix domain facilitates the accumulation of the Gag protein at the plasma membrane [5]. The capsid and NC proteins promote Gag-Gag interactions during virus assembly [6,7]. Additionally, the C-terminal p6 protein binds several viral accessory proteins and facilitates virus release.

The time course of HIV-1 assembly and release has been investigated in living cells within the past two decades. However, the spatiotemporal dynamics of Gag protein trafficking from its synthesis site in the cytoplasm to the budding site remain poorly understood. Transmission electron microscope (TEM) has been a widely used method for studying HIV-1 assembly and release. However, TEM is not suitable for dynamically observing and analyzing the virus assembly and release process on the cell membrane. To address this limitation, Helma et al. developed a high-affinity single-domain antibody, also known as a nanobody, which specifically recognizes the HIV-1 capsid protein [8]. This nanobody was conjugated with a fluorescent protein to enable the detection of HIV-1 morphogenesis in living cells. In this study, the pcHIV and CANTDcb1eGFP plasmids were co-transfected into HeLa-Kyoto cells, and confocal spinning disk microscopy was employed for 90 minutes at 1-minute intervals to visualize HIV-1 formation. The data revealed that the HIV-1 assembly process primarily occurs on the cell membrane region. However, the detailed dynamics of virus assembly and release remain unclear, as the data were displayed at 10-minute intervals over the 90-minute period.

It is well established that the Gag protein plays a critical role in the late stages of the HIV-1 replication cycle [9–11]. Mediated by the NC domain, the HIV-1 Gag protein binds to viral genomic RNA (gRNA), acting as a scaffold that induces the formation of small Gag oligomers in the cytoplasm [12–14]. Subsequently, the Gag-gRNA complex is targeted to the plasma membrane in T lymphocytes as well as in non-physiological cell lines, such as HEK293T, HeLa, and COS cells [15]. Membrane binding is facilitated by the myristoylation of the MA domain at the N-terminus and a highly basic

region (HBR) on the MA surface [16,17]. Interestingly, Mouland A. recently demonstrates that Gag promotes the formation of cytoplasmic condensates involved in viral assembly [18], in addition Di Nunzio F. showed that the processed form of Gag, the viral capsid, facilitates the formation of nuclear viral condensates (HIV-1 membraneless organelles), which are associated with post-nuclear entry events [19,20].

Accumulating evidence suggests that Gag-GFP fusions and native Gag proteins exhibit similar functionalities [21]. Gag expression alone has been observed to generate virus-like particles (VLPs) or virological synapses in various human cell types [13,22]. The dynamics of Gag-GFP fusion proteins provide a valuable model for studying the formation of HIV-1 particles [23]. Studies have demonstrated that Gag proteins can bind to the cell membrane within 5–10 minutes post-synthesis [24,25]. Subsequently, Gag undergoes higher-order multimerization, forming a spherical, immature capsid shell that encapsulates the viral RNA genome [15]. Additionally, research indicates that Gag proteins can also assemble at late endosomal/multivesicular body (LE/MVB) compartments [26].

In this study, we engineered a Gag-EGFP fusion protein that can be directly detected using LCM. Subsequently, the "xyt" and "xyz" recording modes of LCM were employed to observe the dynamic and spatial distribution of the Gag-EGFP fluorescent protein at the plasma membrane and within the cytoplasm, respectively.

## Materials and methods

### Plasmid constructs

The HIV p55 cDNA was amplified via PCR from the pLP1 vector (Invitrogen) and subsequently cloned into the pEGFP-N3 vector using Xho I and BamH I restriction enzymes. The successful cloning was confirmed through sequencing.

### Cell culture

HEK293T cells were cultured in DMEM (Hyclone) supplemented with 10% fetal bovine serum, 4.5 g/L glucose, 4.0 mM L-glutamine, and sodium pyruvate. The cells were incubated at 37°C in a humidified atmosphere containing 5% $CO_2$.

### DNA transfection

For transfection, HEK293T cells ($1.5 \times 10^5$) were seeded into 35 mm confocal dishes (NEST). On the day of transfection, the cells had reached 50% to 60% confluency. HEK293T cells were transfected with pEGFP-N3 vectors or pEGFP-N3-Gag vectors (3 µg/well). Transfection was performed using Lipofectamine 2000 (Invitrogen) following the manufacturer's protocol.

### Laser confocal microscopy detection and analysis

Twenty-four hours post-transfection, the cells were analyzed using a Leica TCS SP5 confocal microscope equipped with Argon 488 nm lasers. Sequential confocal scanning was conducted to capture images, and the dynamic distribution of Gag-EGFP fluorescent protein at the plasma membrane or in the cytoplasm was observed using the "xyt" mode. The motion trajectory of GCC in living cells was manually tracked using the MtrackJ plugin in ImageJ 1.54p software. Subsequently, multiple motion trajectory images were converted into dynamic images using Easy GIF Animator 7.3 software at a rate of 2 frames per second. Additionally, Z-stacks were acquired in xyz mode using LCM, and images were processed with the Volume Viewer plugin in ImageJ 1.54p software to perform 3D reconstruction of target proteins.

### Statistical analysis

Statistical analyses were conducted using an unpaired Student's t-test in GraphPad Prism 9 (San Diego, USA). Each experiment was replicated at least three times, and the data are presented as means ± standard deviation (SD). Differences were considered statistically significant when the p-value was less than 0.05.

# Result

## Real-time observation of the motion characteristics of gag proteins in living Cells

As depicted in Fig 1A, the movement trajectories of three Gag-containing complexes (GCC) localized on the cell membrane were recorded. The positions of the three GCC exhibited continuous changes over a short period, with GCC 1 and 3 displaying smaller displacement ranges, while GCC 2 exhibited a larger displacement range (S1 Movie).

Additionally, Gag-EGFP proteins were observed to form GCC in the cytoplasm (Fig 1B), and these complexes underwent a defined range of motion (S2 Movie). Initially, the first and second GCC in the cytoplasm were separated by a certain distance (10 s). As they moved, the two complexes gradually came into contact (50 s) and partially merged (100 s). Complete fusion was observed at 150 s, and this fusion state was maintained for a sustained period (180 s).

## Dynamic observation of GCC detaching from the cell membrane

As illustrated in Fig 2A (S3 Movie), small circular GCC initially exhibited small-scale movement on the cell membrane (10 s, 50 s, 100 s, 120 s, 180 s) and subsequently detached from the membrane (200 s). As the GCC moved freely, they gradually disappeared from the observation field (250 s, 290 s). Simultaneously, larger Gag particle clusters on the cell membrane were also observed to release via budding (S4-S5 Movie). A larger particle, initially distributed on the cell membrane in a wedge-shaped configuration (0 min), was released through free swinging (3 min, 8 min). The detached large particle retained certain movement characteristics, causing it to continuously shift away from the cell (49 min, 51 min).

## The spatial distribution characteristics of GCC on the cell membrane

To further elucidate the spatial distribution characteristics of Gag proteins on the cell membrane, 60 consecutive cell section images were captured using LCM in xyz scanning mode. In the first scanned image (near the glass surface), Gag-EGFP proteins were concentrated in two distinct regions of the cell membrane (Fig 3E1). At the same time, the 60th

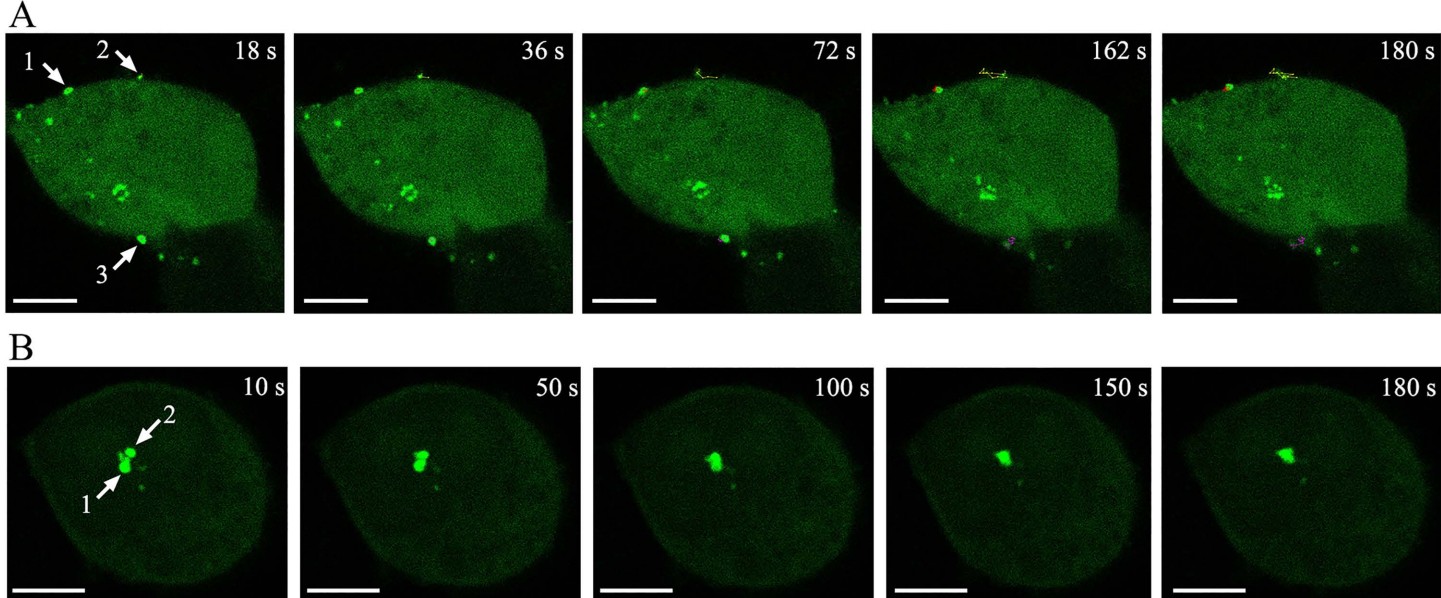

**Fig 1. Dynamic imaging of motion characteristics of Gag-EGFP proteins in living cells.** HEK293T cells were transfected with pEGFP-N3-Gag, and dynamic imaging was conducted 24 hours post-transfection using the xyt mode of laser confocal microscopy (LCM). (A) Real-time trafficking of the GCC trajectory on the cell membrane. Scale bars = 5 µm. (B) Real-time trafficking of the GCC trajectory in the cytoplasm. Scale bars = 5 µm.

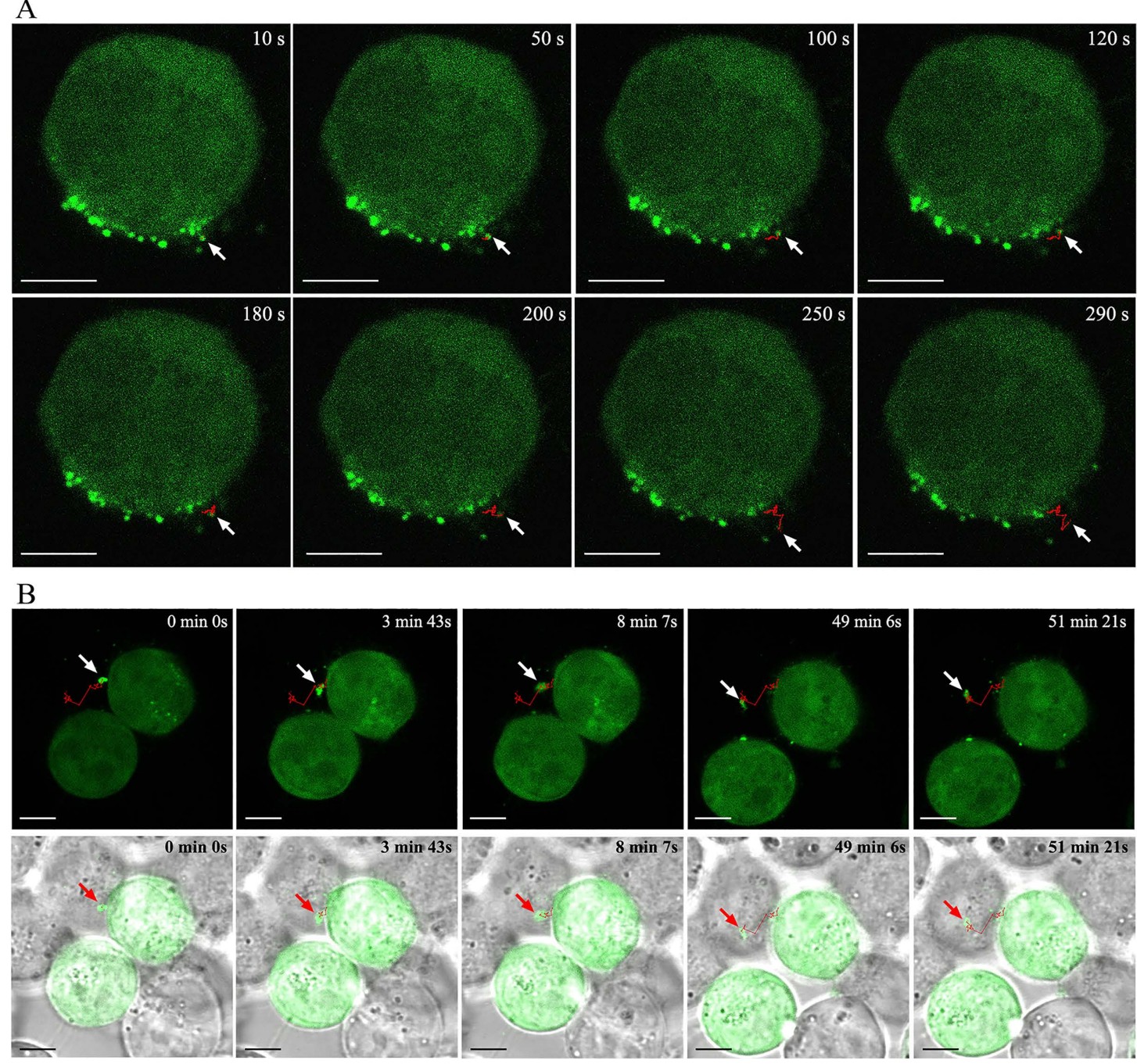

**Fig 2. Real-time observation of GCC budding from living cells.** HEK293T cells were transfected with pEGFP-N3-Gag, and dynamic imaging was performed using the xyt mode of LCM 24 hours post-transfection. (A) GCC detached from cells within a short time frame (approximately 3 minutes). Scale bars = 5 μm. (B) Larger GCC detached from cells over a longer period (approximately 8 minutes). Scale bars = 5 μm.

scanned image (away from the glass surface) revealed a more dispersed distribution of Gag-EGFP proteins, with some appearing as small circular dots (Fig 3E2). The 3D reconstruction results demonstrated that Gag-EGFP proteins can exist in three forms: granule like structures, cord like structures, and giant polymers. The maximum longitudinal diameter of

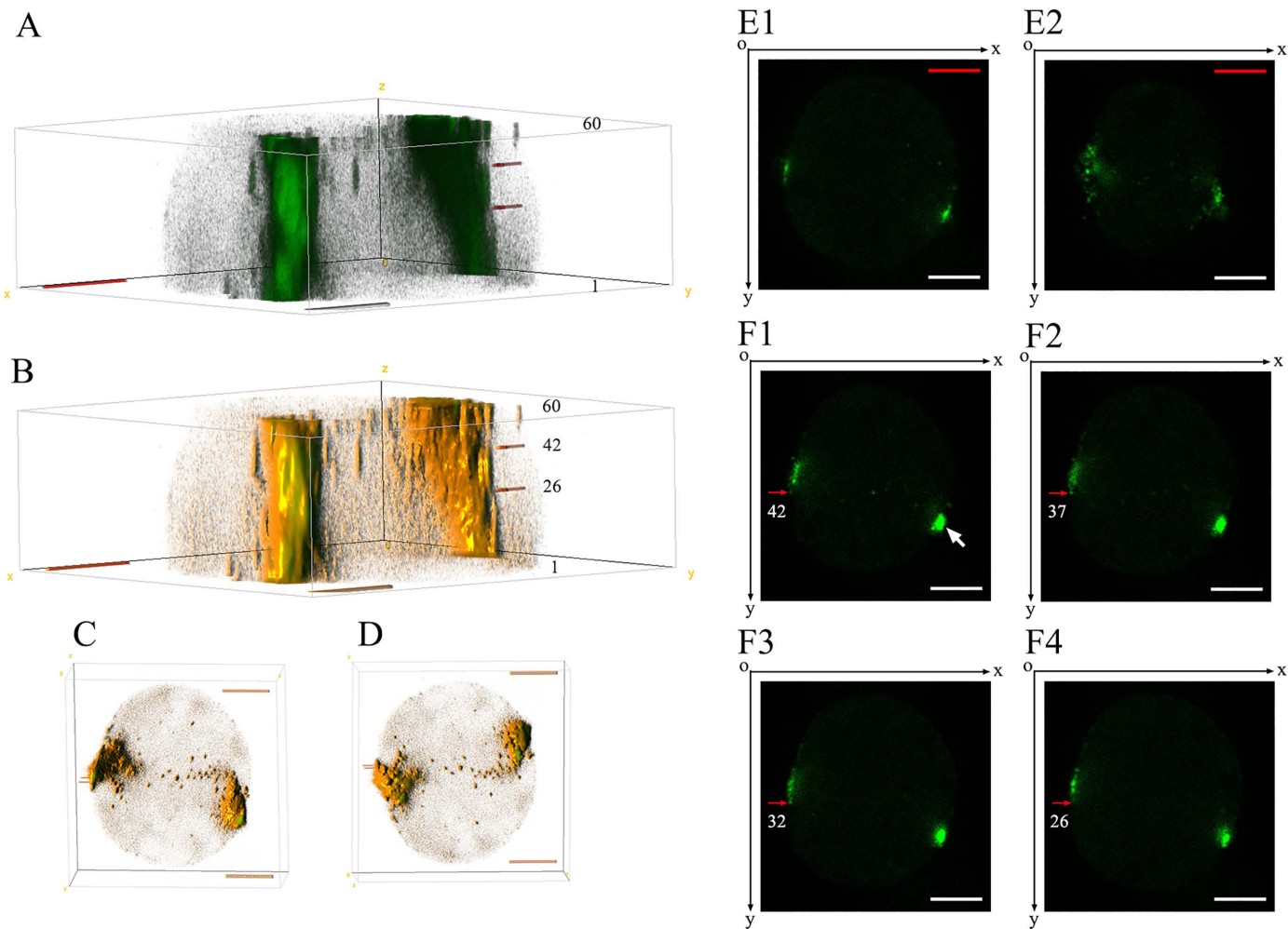

**Fig 3. 3D reconstruction to identify spatial distribution characteristics of GCC on the cell membrane.** 3D reconstruction was employed to identify the spatial distribution characteristics of GCC on the cell membrane. HEK293T cells were transfected with pEGFP-N3-Gag, and imaging was conducted using the xyz mode of LCM 24 hours post-transfection. Scale bars = 5 μm. (A-D) A 3D image of GCC on the cell membrane is presented, with 60 horizontal slices shown from the bottom to the top of the sample. (A) Side view. (B) Side view (Light rendering effects added). (C) Top view. (D) Bottom view. (E1–E2) The first and 60th horizontal slices illustrating the GCC on the cell membrane are displayed. (F1–F4) The 42nd, 37th, 32nd, and 26th horizontal slices illustrating the GCC on the cell membrane are depicted.

these aggregates reached 7.67 μm (with a distance of 0.13 μm between consecutive layers, totaling 60 layers), and the maximum transverse diameter was 2 μm (Fig 3A-3D).

As shown in Fig 3F1–F4, fluorescent protein spots (indicated by red arrows) were distributed along the meridian direction of the cell surface, corresponding to the 26th, 32nd, 37th, and 42nd cell sections. The 3D reconstruction further revealed that these Gag-EGFP proteins (indicated by red arrows) aggregated into long, fine strands distributed between layers 26–42, with a longitudinal diameter of approximately 2.08 μm and a maximum transverse diameter of 0.36 μm (Fig 3B). Interestingly, we found that the long axis of these elongated aggregates were aligned closely with the Z-axis direction.

## Spatial distribution characteristics of GCC in the cytoplasm

Following the expression of Gag-EGFP proteins, they primarily migrated to the cell membrane. Concurrently, some Gag-EGFP proteins bound to each other in the cytoplasm, forming larger cytoplasmic aggregates. To further elucidate the

spatial distribution characteristics of Gag-EGFP proteins in the cytoplasm, LCM images were continuously captured using xyz scanning mode (layers 1–25). In the first layer of the LCM scan, Gag-EGFP proteins formed a single circular polymer (Fig 4B2). In subsequent scanning sections (e.g., the 10th layer), Gag-EGFP proteins were observed to form three circular aggregates of varying sizes (Fig 4B3).

The 3D reconstruction results revealed that Gag polymers can exist in two forms in the cytoplasm: granule like structures and cord like polymers (Fig 4A). Among these, the third polymer exhibited the smallest volume, with its main body appearing in the 10th, 16th, and 20th layers of the confocal image (Fig 4B1, No. 3). Its longitudinal axis measured approximately 1.3 µm (between the 10th and 20th layers), and its maximum transverse diameter was about 0.38 µm (16th layer). All three aggregates exhibited an elongated shape, with their longitudinal axes aligned with the Z-axis direction.

## Discussion

In this study, we discovered that HIV-1 Gag proteins can form complexes either on the dorsal cell membrane or in the cytoplasm, and these complexes exhibit varying degrees of motility. Some Gag complexes (GCC) display the attribute of rapid movement with a broad range of activity, while others exhibit slower speeds and more confined movement (Fig 1). Our findings align with those of Manley et al., who successfully visualized Gag proteins in live COS7 cells by integrating photoactivated localization microscopy (PALM) with single-particle tracking [27]. Specifically, a 405 nm excitation light was employed to randomly activate a subset of Gag-tdEos fluorescent proteins, followed by activation using a 561 nm laser. After data collection, these fluorescent proteins were photobleached to complete inactivation, allowing for the activation of

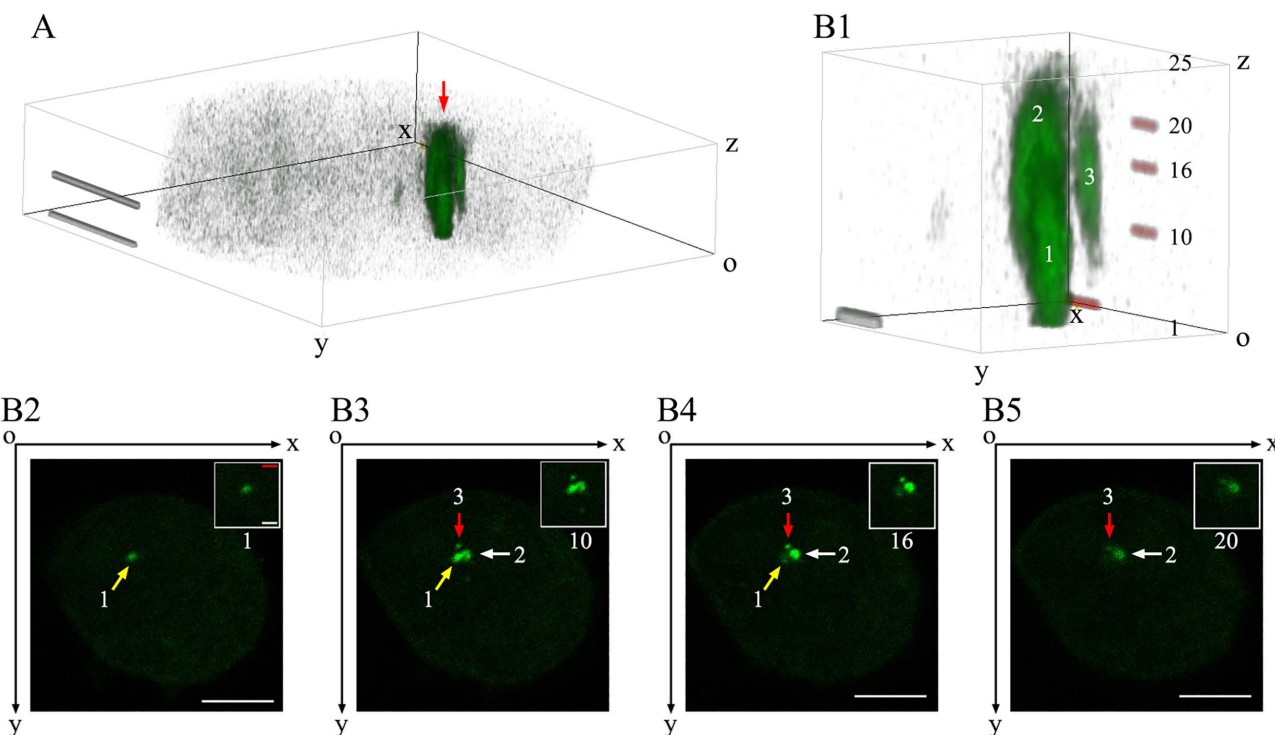

**Fig 4. 3D reconstruction to identify spatial distribution characteristics of GCC in the cytoplasm.** HEK293T cells were transfected with pEGFP-N3-Gag, and imaging was conducted using the xyz mode of LCM. Scale bars = 5 µm. (A) A 3D image of GCC in the cytoplasm is presented, with the 3D reconstruction performed on 24 images using the Volume Viewer 2.0 plugin of ImageJ software. The result is shown from a lateral upper perspective. (B1) The 3D reconstruction of intracellular Gag polymer. (Display after zooming in, indicated by the red arrow in Fig 1A). (B2–B5) The first, 10th, 16th, and 20th horizontal slices illustrating the GCC in the cytoplasm are displayed.

other fluorescent proteins for localization. By iterating this process, all target molecules were localized, and merging these raw data yielded a super-resolution image of the target molecules. In the present work, mobile Gag-Eos molecules were observed with path lengths of up to ~1 µm [27].

Secondly, we observed that GCC particles of varying sizes on the dorsal cell membrane can be released by budding within 3–8 minutes. Approximately 20 years ago, classic methods were employed to visualize the HIV-1 life cycle in live cells [28]. Total internal reflection fluorescence microscopy (TIRFM) has been widely used for the dynamic observation of HIV-1 assembly and release. The detection range of TIRFM is typically limited to within 200 nm above the glass interface (Fig 5-8). Due to the exponential decay of excitation light, only the sample area in close proximity to the total reflection surface produces fluorescence. Ivanchenko et al. demonstrated that the Gag protein rapidly aggregates at the budding site, completing approximately 90% of the virus assembly within 8–9 minutes [29]. After nucleation in the assembly area, the virus release occurred in approximately 1,500 ± 700 seconds. However, the narrow space between the cell membrane of adherent cells and the glass medium surface, referred to as the ventral membrane (Fig 5-5), warrants careful consideration regarding its potential impact on HIV-1 assembly and release [23,30]. In contrast, the dorsal membrane faces a more natural and open extracellular space, which may provide a more suitable environment for observing virus assembly and release (Fig 5-4). Using a correlated scanning electron microscopy (SEM) and multiphoton fluorescence microscopy approach, Larson et al. observed the budding of virus-like particles of RSV and HIV-1 [31]. Through real-time imaging of RSV budding, they determined that some representative particles took approximately 14 minutes to disappear from subsequent frames.

Thirdly, we observed that the Gag-EGFP proteins formed numerous elongated, rope-like aggregates on the cell membrane or within the cytoplasm in HEK293T cells transfected with the pEGFP-N3-Gag vectors (Fig 3A and 4A). It is crucial to note that the small spot structures observed through LCM at a single focal plane may exhibit significantly larger actual

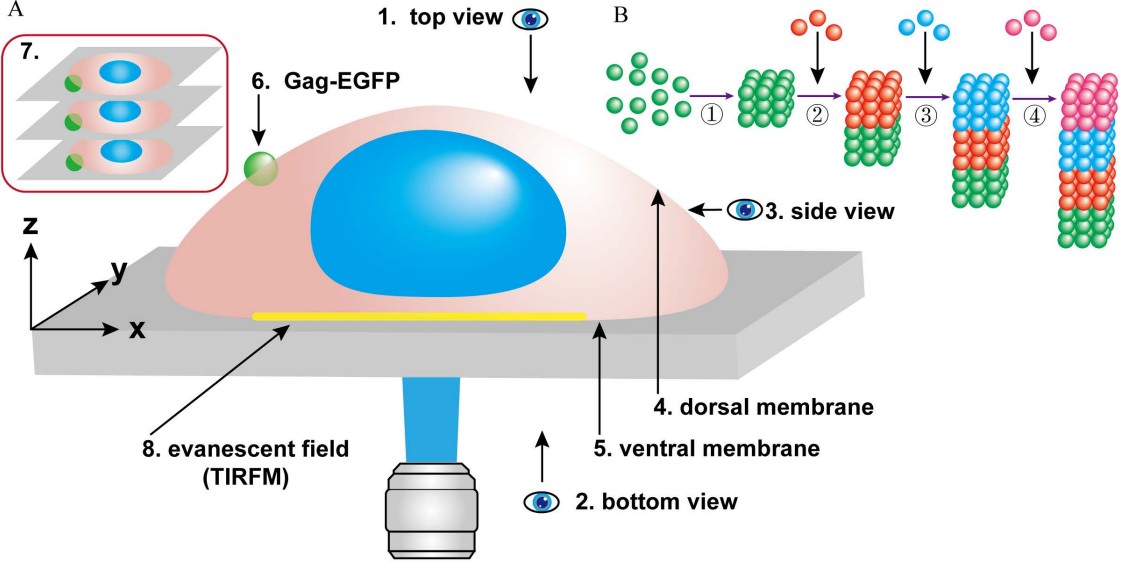

**Fig 5. A schematic illustration demonstrates the spatial distribution of Gag-EGFP proteins as visualized through 3D reconstruction.** (A) Schematic representation of the 3D reconstruction simulation: (1) top view, (2) bottom view, (3) side view, (4) dorsal membrane, (5) ventral membrane, and (6) Gag-EGFP proteins distributed on the cell membrane. (7) Overlaying multiple layers of confocal images for 3D reconstruction. (8) Evanescent field for TIRFM detection. **(B)** The prediction model to illustrate the formation of a linear protein polymer: (1) Multiple Gag-EGFP proteins bind to form a polymer. (2) Additional Gag-EGFP proteins aggregate and bind to the backside of the existing polymer, causing it to move downward due to gravitational force. (3) and (4) Gag-EGFP proteins continue to polymerize and sink, extending the polymer length in the Z-axis direction.

sizes and unexpected shapes within the cell. Meanwhile, we found that the long axis of these aggregates aligned with the Z-axis direction, approximately perpendicular to the bottom of the confocal glass slide. This phenomenon implies that the shape of the aggregates might be influenced by the resultant force in the Z-axis direction during their formation. We proposed the following model: when multiple Gag protein molecules assembled into a macromolecular polymer on the cell membrane, they probably underwent sedimentation under the influence of the resultant force in the Z-axis direction (Fig 5B). Consequently, new Gag protein molecules continued to polymerize on the underside of the macromolecular polymer. Through repeated cycles of this process, the polymer formed a linear structure (Fig 5B). This prediction model is somewhat similar to the report by Feric M [32]. The diameter of specific cells (oocytes) or nuclei can exceed 10 µm, whereas the diameter of their nucleoli and histone locus bodies (HLBs) can reach up to 1 µm. Under such circumstances, gravitational forces can lead to the sedimentation of the nucleolus and HLBs. However, this sedimentation can be counteracted by actin molecules. In this study, the distribution regions of Gag proteins on the cell membrane or within the cell ranged approximately from 1 to 2 µm (S1 Fig). Nevertheless, there was no significant statistical difference in the length of the Gag proteins between the two groups (P > 0.05). In S2 Fig, we found that the distribution of GCC on the dorsal cell membrane significantly exceeded its distribution inside the cell (P < 0.01). At the same time, the quantity of GCC on the cell membrane or inside the cell exhibits certain variations at different observation levels (data not shown). Whether gravity is involved in the formation of GCC and what the main forces leading to the GCC formation are remain questions that need to be further clarified in the follow-up study.

When dynamically observing the assembly and release of HIV virus, several issues require sufficient attention from observers. Dynamic observation can more objectively and accurately reflect the process of HIV virus assembly and release, avoiding some one-sided or erroneous interpretation of results. It is worth noting that the fusion of the two Gag-containing complexes (Fig 1B), triggered by movement, may be genuine or pseudo. In a previous study, HIV Gag-iGFP ΔEnv plasmids were transfected into macrophages, and spinning disk confocal microscopy was employed to acquire 3D images every 5 minutes for two hours. From a top view, two HIV-containing compartments were observed to fuse within 30 minutes [33]. In our experiment, a similar result was initially obtained. Within the observation field selected in Fig 1B, we used the xyt mode to capture images and observed the apparent fusion of the two complexes. However, when we subsequently used the xyz mode to recapture images and performed 3D reconstruction using ImageJ software, we found that the two complexes did not truly fuse and remained as independent entities (Fig 4B1, 4B3). This discrepancy can be attributed to one of the complexes moving out of the observation field of view. When GCC1 experiences a resultant force along the Z-axis, it moves downward, causing it to no longer lie in the same observation plane as GCC2 (Fig 4B1).

In previous studies, researchers often used the disappearance of specific spots within a single frame as the criterion for identifying a budding event [23]. Contrary to this perspective, our results revealed that certain particles disappeared from the observed image at a specific time point but later reappeared in the observation field. This phenomenon may be attributed to their movement along the Z-axis. Meanwhile, some scholars have argued that the disappearance of fluorescent spots in an observation field can only serve as circumstantial evidence for such events [31]. A more convincing criterion is the disappearance of fluorescent spots across multiple consecutive sequence scanning images [31].

Moreover, compared with the ventral cell membrane, analyzing the assembly and release of HIV virus using the dorsal cell membrane as the observation object may be closer to the real process. In recent years, the emergence of novel detection methods has provided new insights into the real-time morphological changes of the HIV assembly and budding process in host cells [34–36]. For instance, a scanning ion conductance microscopy and fluorescence confocal microscopy (SICM-FCM) system, which employs a non-contact scanning probe, has been utilized to detect the three-dimensional (3D) morphology of cells with high resolution [23]. Bednarska et al. demonstrated that most assembled HIV virus-like particles are released from HEK293T and Jurkat cells within 0.5 to 3 minutes [23]. A significant advantage of SICM-FCM is its ability to observe morphological changes on the dorsal membrane of cells (Fig 5−4). In this study, we found that laser confocal microscopy can be effectively used for detection the assembly and release of HIV virus on the

dorsal cell membrane. In contrast, many previous studies primarily focused on the ventral membrane, which constitutes the bottom layer of the detection surface (Fig 5-5).

Furthermore, the time interval for image acquisition should be set reasonably when dynamically observing the assembly and release of HIV virus. If the time interval is too small, it will accelerate the quenching of the fluorescence signal, and the data volume will be excessively large. On the other hand, excessive time intervals can lead to the loss of certain valuable information. To dynamically observe Gag trafficking in physiologically relevant cell types, Gousset et al. inserted a small tetracysteine tag after the matrix domain of Gag, which is specifically recognized by the membrane-permeable biarsenical dyes FlAsH and ReAsH [37]. At multiple specific time points (0, 10, 18, 27, 36, 61, and 81 minutes), no significant movement of Gag was detected at the plasma membrane or in the cytoplasm of living primary monocyte-derived macrophages (MDMs) infected with VSV-G-pseudotyped NL4–3/MA-TC virus. These results differ significantly from our findings, which may be attributed to several factors. First, Gousset et al. used MDM cells, whereas we employed HEK293T cells. Second, their study utilized a fluorescence microscope (Olympus IX-71) for image acquisition, while we used a laser confocal microscope that was not easy to cause sample fluorescence quenching. Third, the time interval for continuous image acquisition in Gousset et al.'s study was approximately 5–10 minutes, compared to 10–30 seconds in our study. The longer time interval in their study may have resulted in the loss of some critical information regarding Gag protein movement.

At the same time, this study still encounters specific limitations. The sole use of Gag over-expression systems without authentic HIV viral infection models may restrict the physiological relevance of the findings. A more comprehensive understanding of the spatiotemporal distribution characteristics of HIV-1 GCC can be achieved through direct authentic viral infection models combined with live imaging.

## Conclusions

HIV-1 Gag proteins exhibited directional migration toward the cell membrane, where they polymerized, displaying localized small-scale movement and the capacity to detach from the membrane at varying rates. Furthermore, these proteins assembled into large polymers or elongated linear structures both on the cell membrane and within the cytoplasm. Although constrained by the diffraction limit, LCM remains a valuable auxiliary tool for studying the assembly and release of HIV-1 virions. By continuously optimizing the scanning parameters of LCM in the xyz or xyt mode, dynamic observation of morphological changes on the dorsal membrane can be readily achieved.

## Supporting information

**S1 Movie. Laser confocal microscope (LCM) imaging of the movement trajectories of Gag-containing complexes (GCC) localized on the cell membrane.**
(GIF)

**S2 Movie. LCM imaging of the movement of GCC localized in the cytoplasm.**
(GIF)

**S3 Movie. LCM imaging of the detaching of small circular GCC from the cell membrane.**
(GIF)

**S4 Movie. LCM imaging of the detaching of larger Gag particle from the cell membrane.**
(GIF)

**S5 Movie. LCM imaging of the detaching of larger Gag particle from the cell membrane (Overlay of bright field image and fluorescence field image).**
(GIF)

**S1 Fig. The distribution of Gag proteins, whether on the cell membrane or inside the cell, varies.**
(TIF)

**S2 Fig. The counts of GCC on the dorsal cell membrane or within the cells.**
(TIF)

## Author contributions

**Conceptualization:** Dakang Sun.

**Data curation:** Xiao Wang, Xinye An.

**Formal analysis:** Xiao Wang, Xinye An.

**Funding acquisition:** Dakang Sun.

**Investigation:** Xiao Wang, Xinye An, Hongmei Zhao.

**Project administration:** Dakang Sun.

**Visualization:** Dakang Sun.

**Writing – original draft:** Dakang Sun.

**Writing – review & editing:** Dakang Sun.

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
