## [Decision Letter · Decision Letter 0]

8 Sep 2025

Dear Dr. Sun,

Thank you for submitting your manuscript to PLOS ONE. After careful consideration, we feel that it has merit but does not fully meet PLOS ONE’s publication criteria as it currently stands. Therefore, we invite you to submit a revised version of the manuscript that addresses the points raised during the review process.

We look forward to receiving your revised manuscript.

Kind regards,

Mauricio Comas-Garcia

Academic Editor

PLOS ONE

Journal Requirements:

This work was supported by by Projects of medical and health technology development program in Shandong province (grant numbers 2018WS545)

The authors declare no competing interests.

Additional Editor Comments:

Reviewer #1:

The study is interesting and uses cutting edge imaging tools to investigate the dynamics of virus assembly and release. I have already revised the article for another journal, while the authors included some comments the study is still not ready to be published, according to my opinion. The study is very short. About the morphology of Gag aggregates that may have been influenced by gravitational forces during their formation I do not see strong evidence in the study. While this is a scientifically appealing idea, it remains speculative.

Additionally, the bibliography appears sometimes cited not correctly. For example, recent work by Andrew Mouland’s group on Gag and condensates could significantly enhance the discussion. Their studies demonstrate that Gag contributes to the formation of cytoplasmic condensates involved in viral assembly, in addition the processed form of Gag giving rise viral cores and these cores during nuclear entry contribute to the formation of nuclear viral condensates (HIV-1 MLOs), which are relevant to post-nuclear entry events (see: Scoca et al., JMCB, 2022; Ay et al., EMBO J., 2024). However, the authors included the following sentence that it is not correct and it should be corrected: ” Interestingly, Nunzio F.D. recently demonstrates that Gag promotes the formation of cytoplasmic condensates involved in viral assembly, in addition the processed form of Gag facilitates the formation of nuclear viral condensates (HIV-1 membraneless organelles), which are associated with post-nuclear entry events[18, 19].” I advise to the authors to change it with the following sentence: Interestingly, Mouland A. recently demonstrates that Gag promotes the formation of cytoplasmic condensates involved in viral assembly, in addition Di Nunzio F. showed that the processed form of Gag, the viral capsid, facilitates the formation of nuclear viral condensates (HIV-1 membraneless organelles), which are associated with post-nuclear entry events[18, 19].

Incorporating these findings could strengthen the contextual understanding of Gag’s role in viral condensate formation.

Finally, I recommend including a dedicated section discussing the limitations of the study. The exclusive use of Gag overexpression systems without authentic viral infection models may limit the physiological relevance of the findings. Addressing this would help contextualise the study’s scope and implications more transparently.

In my opinion, the imaging methods used by the authors are adequate. Even though some of the conclusions—particularly those regarding the role of gravity force on aggregate formation—require further investigation and should be presented as hypotheses, if the authors agree to include a discussion of the limitations of their study, I believe the manuscript would be suitable for publication in PLOS ONE.

Reviewer #2:

This study uses a laser confocal microscope to observe Gag (GCC) complex trafficking in cells in real time. The manuscript is well-written, and the data is solid. While it is in good form for acceptance, it could benefit from minor revisions.

To enhance the work, the authors could perform statistical analysis on the localization results. Currently, the manuscript estimates the GCC location at either the dorsal membrane or cytosol and measures distances with micrometer accuracy. It would be helpful to also analyze statistically: (1) the proportion of GCC at each location, including the percentage of events at each site; and (2) the distribution of particles within a specific location or image layer, (3) for the distance measurement, what is the standard deviation?

Reviewers' comments:

Reviewer's Responses to Questions

**Comments to the Author**

1. Is the manuscript technically sound, and do the data support the conclusions?

Reviewer #1: Partly

Reviewer #2: Yes

2. Has the statistical analysis been performed appropriately and rigorously?

Reviewer #1: I Don't Know

Reviewer #2: N/A

3. Have the authors made all data underlying the findings in their manuscript fully available?

Reviewer #1: Yes

Reviewer #2: Yes

4. Is the manuscript presented in an intelligible fashion and written in standard English?

Reviewer #1: Yes

Reviewer #2: Yes

Reviewer #1: The study is interesting and uses cutting edge imaging tools to investigate the dynamics of virus assembly and release. I have already revised the article for another journal, while the authors included some comments the study is still not ready to be published, according to my opinion. The study is very short. About the morphology of Gag aggregates that may have been influenced by gravitational forces during their formation I do not see strong evidence in the study. While this is a scientifically appealing idea, it remains speculative.

Additionally, the bibliography appears sometimes cited not correctly. For example, recent work by Andrew Mouland’s group on Gag and condensates could significantly enhance the discussion. Their studies demonstrate that Gag contributes to the formation of cytoplasmic condensates involved in viral assembly, in addition the processed form of Gag giving rise viral cores and these cores during nuclear entry contribute to the formation of nuclear viral condensates (HIV-1 MLOs), which are relevant to post-nuclear entry events (see: Scoca et al., JMCB, 2022; Ay et al., EMBO J., 2024). However, the authors included the following sentence that it is not correct and it should be corrected: ” Interestingly, Nunzio F.D. recently demonstrates that Gag promotes the formation of cytoplasmic condensates involved in viral assembly, in addition the processed form of Gag facilitates the formation of nuclear viral condensates (HIV-1 membraneless organelles), which are associated with post-nuclear entry events[18, 19].” I advise to the authors to change it with the following sentence: Interestingly, Mouland A. recently demonstrates that Gag promotes the formation of cytoplasmic condensates involved in viral assembly, in addition Di Nunzio F. showed that the processed form of Gag, the viral capsid, facilitates the formation of nuclear viral condensates (HIV-1 membraneless organelles), which are associated with post-nuclear entry events[18, 19].

Incorporating these findings could strengthen the contextual understanding of Gag’s role in viral condensate formation.

Finally, I recommend including a dedicated section discussing the limitations of the study. The exclusive use of Gag overexpression systems without authentic viral infection models may limit the physiological relevance of the findings. Addressing this would help contextualise the study’s scope and implications more transparently.

Reviewer #2: This study uses a laser confocal microscope to observe Gag (GCC) complex trafficking in cells in real time. The manuscript is well-written, and the data is solid. While it is in good form for acceptance, it could benefit from minor revisions.

To enhance the work, the authors could perform statistical analysis on the localization results. Currently, the manuscript estimates the GCC location at either the dorsal membrane or cytosol and measures distances with micrometer accuracy. It would be helpful to also analyze statistically: (1) the proportion of GCC at each location, including the percentage of events at each site; and (2) the distribution of particles within a specific location or image layer, (3) for the distance measurement, what is the standard deviation?

**Do you want your identity to be public for this peer review?** For information about this choice, including consent withdrawal, please see our Privacy Policy

Reviewer #1: No

Reviewer #2: No

---

## [Author Response · Author response to Decision Letter 1]

15 Oct 2025

Dear Editor and Reviewers:

Thank you for your letter and the reviewers’ comments regarding our manuscript titled “Characteristics of spatiotemporal distribution of HIV-1 Gag-containing complexes on the dorsal membrane tracking with live confocal imaging” (ID: PONE-D-25-44222).

These comments are valuable and extremely helpful for revising and enhancing our paper. They also offer significant guidance for our research. We have carefully studied the comments and made corrections, which we hope will be approved by you. The revised sections in the paper are marked in red. Below are the main corrections in the paper and our responses to the reviewers’ comments.

Editors

We sincerely appreciate your valuable feedback, which we have utilized to enhance the quality of our manuscript. Based on your valuable suggestions, all descriptive terms related to gravity have been removed from the experimental results. In line with your comments, we have made the necessary corrections.

1. Response to comment: (Please ensure that your manuscript meets PLOS ONE's style requirements, including those for file naming. The PLOS ONE style templates can be found at https://journals.plos.org/plosone/s/file?id=wjVg/PLOSOne_formatting_sample_main_body.pdf and https://journals.plos.org/plosone/s/file?id=ba62/PLOSOne_formatting_sample_title_authors _affiliations.pdf)

(1) We have carefully revised the article format to meet the style requirements of PLOS ONE's. The primary reference documents are the following two files. (PLOS ONE formatting sample main body.pdf, PLOS ONE formatting sample title authors affiliations.pdf). Below are the main corrections in the paper.

1) Adjustments have been made to the content of Author Byline Affiliations Corresponding Authorship Contributorship.

2) Replace all “Fig.” in the entire text with “Fig”.

3) Delete the additional information text content before the supplementary information.

4) Moving the Supporting Information module forward to the front of acknowledgements.

5) Referring to the published articles in PLOS ONE, adjustments have been made to the content of the Author Contributions module.

(2) Referring to the following 3 references in PLOS ONE, we have made corresponding adjustments to the fonts and font sizes of each module throughout the entire text.

1) D. Bekric, T. Kiesslich, M. Ocker, M. Winklmayr, M. Ritter, H. Dobias, M. Beyreis, D. Neureiter, C. Mayr, The efficacy of ferroptosis-inducing compounds IKE and RSL3 correlates with the expression of ferroptotic pathway regulators CD71 and SLC7A11 in biliary tract cancer cells, PLoS One 19 (2024) e0302050.

2) J. Lei, W. Chen, Y. Gu, X. Lv, X. Kang, X. Jiang, Ferroptosis regulation by traditional chinese medicine for ischemic stroke intervention based on network pharmacology and data mining, PLoS One 20 (2025) e0321751.

3) Y. Yokomaku, T. Noda, M. Imahashi, Y. Nishioka, T. Myojin, A. Iwamoto, T. Imamura, Antiretroviral therapies and status of people living with HIV in Japan: An update from hospital survey and national database, PLoS One 20 (2025) e0317655.

2. Response to comment: (We note that the grant information you provided in the ‘Funding Information’ and ‘Financial Disclosure’ sections do not match. When you resubmit, please ensure that you provide the correct grant numbers for the awards you received for your study in the ‘Funding Information’ section.)

We provide the correct grant numbers in the ‘Funding Information’ section. The funding information has been placed in the cover letter.

Funding: This work was supported by Projects of medical and health technology development program in Shandong province (Grant 2018WS545 to Dakang Sun). The funders had no role in study design, data collection and analysis, decision to publish, or preparation of the manuscript.

3. Response to comment: (Thank you for stating the following financial disclosure:

This work was supported by Projects of medical and health technology development program in Shandong province (grant numbers 2018WS545)

Please include this amended Role of Funder statement in your cover letter; we will change the online submission form on your behalf.)

The information content of 'Financial Disclosure' has been updated as follows. The funding information is as follows and has been placed in the cover letter.

Funding: This work was supported by Projects of medical and health technology development program in Shandong province (Grant 2018WS545 to Dakang Sun). The funders had no role in study design, data collection and analysis, decision to publish, or preparation of the manuscript.

4. Response to comment: (Thank you for stating the following in the Competing Interests section:

The authors declare no competing interests.

Please include your updated Competing Interests statement in your cover letter; we will change the online submission form on your behalf.).

The updated information content of 'Competing interests ' is as follows and has been placed in the cover letter.

Competing interests: The authors have declared that no competing interests exist. This does not alter our adherence to PLOS ONE policies on sharing data and materials.

5. Response to comment: (If the reviewer comments include a recommendation to cite specific previously published works, please review and evaluate these publications to determine whether they are relevant and should be cited. There is no requirement to cite these works unless the editor has indicated otherwise. )

According to the actual needs of the introduction or discussion, we have added one more reference. The information of a new reference is as follows.

18. Monette A, Niu M, Maldonado RK, Chang J, Lambert GS, Flanagan JM, et al. Influence of HIV-1 Genomic RNA on the Formation of Gag Biomolecular Condensates. Journal of molecular biology. 2023;435(16):168190.

6. Response to comment: (Please review your reference list to ensure that it is complete and correct. If you have cited papers that have been retracted, please include the rationale for doing so in the manuscript text, or remove these references and replace them with relevant current references. Any changes to the reference list should be mentioned in the rebuttal letter that accompanies your revised manuscript. If you need to cite a retracted article, indicate the article’s retracted status in the References list and also include a citation and full reference for the retraction notice.)

We have also reviewed each cited reference to ensure its completeness and accuracy. We have re downloaded the full texts of each reference and have not found any papers that have been retracted.

Reviewer #1:

We sincerely appreciate your professional review of our article. As you have pointed out, there are some issues that need to be addressed. In accordance with your suggestions, we have made extensive revisions to our previous draft. The detailed corrections are listed below.

(1) Response to comment: (About the morphology of Gag aggregates that may have been influenced by gravitational forces during their formation I do not see strong evidence in the study. While this is a scientifically appealing idea, it remains speculative.)

Thanks for your suggestion. All descriptive terms related to gravity have been removed from the experimental results.

(2) Response to comment: (The bibliography appears sometimes cited not correctly. For example, recent work by Andrew Mouland’s group on Gag and condensates could significantly enhance the discussion. Their studies demonstrate that Gag contributes to the formation of cytoplasmic condensates involved in viral assembly, in addition the processed form of Gag giving rise viral cores and these cores during nuclear entry contribute to the formation of nuclear viral condensates (HIV-1 MLOs), which are relevant to post-nuclear entry events (see: Scoca et al., JMCB, 2022; Ay et al., EMBO J., 2024). However, the authors included the following sentence that it is not correct.)

We carefully reviewed the literature and added more references to the revised manuscript. The information of new references are as follows.

Interestingly, Mouland A. recently demonstrates that Gag promotes the formation of cytoplasmic condensates involved in viral assembly[18], in addition Di Nunzio F. showed that the processed form of Gag, the viral capsid, facilitates the formation of nuclear viral condensates (HIV-1 membraneless organelles), which are associated with post-nuclear entry events[19, 20].

18. Monette A, Niu M, Maldonado RK, Chang J, Lambert GS, Flanagan JM, et al. Influence of HIV-1 Genomic RNA on the Formation of Gag Biomolecular Condensates. Journal of molecular biology. 2023;435(16):168190.

19. Scoca V, Morin R, Collard M, Tinevez JY, Di Nunzio F. HIV-induced membraneless organelles orchestrate post-nuclear entry steps. J Mol Cell Biol. 2023;14(11).

20. Ay S, Burlaud-Gaillard J, Gazi A, Tatirovsky Y, Cuche C, Diana JS, et al. In vivo HIV-1 nuclear condensates safeguard against cGAS and license reverse transcription. EMBO J. 2025;44(1):166-99.

(3) Response to comment: (Finally, I recommend including a dedicated section discussing the limitations of the study.)

In the last paragraph of the discussion section, we added the limitations of this study and proposed the feasible solutions to enhance the study’s significance. The relevant additions are as follows.

At the same time,, this study still encounters specific limitations. The sole use of Gag over-expression systems without authentic HIV viral infection models may restrict the physiological relevance of the findings. A more comprehensive understanding of the spatiotemporal distribution characteristics of HIV-1 GCC can be achieved through direct authentic viral infection models combined with live imaging.

Reviewer #2:

We sincerely appreciate your constructive suggestions. In line with your comments, we have made the requisite corrections.

(1) Response to comment: (Currently, the manuscript estimates the GCC location at either the dorsal membrane or cytosol and measures distances with micrometer accuracy. It would be helpful to also analyze statistically. The proportion of GCC at each location, including the percentage of events at each site�)

In Fig A, we found that the distribution of GCC on the dorsal cell membrane significantly exceeded its distribution inside the cell (P<0.01).

Fig A. HEK293T cells were transfected with pEGFP-N3-Gag. Twenty-four hours after transfection, imaging was performed using laser confocal microscopy. Ten fields of view were randomly selected and imaged with a 63x oil immersion objective lens. A count and statistical analysis of the distribution of GCC on the cell membrane or within the cells were carried out.

(2) Response to comment: ( the distribution of particles within a specific location or image layer? )

As shown in Figure B, the quantity of GCC on the cell membrane or inside the cell exhibits certain variations at different observation levels.

Fig B. HEK293T cells were transfected with pEGFP-N3-Gag. Twenty-four hours post-transfection, imaging was conducted using laser confocal microscopy. We randomly selected three adjacent observation layers that could clearly display the distribution of GCC and examined them with a 63x oil immersion objective lens. A count analysis of the distribution of GCC on the cell membrane or within the cells was carried out.

(3) Response to comment: (for the distance measurement, what is the standard deviation?)

In Figure 3B of the initial submission, we randomly selected six thin, elongated, cord-like Gag proteins either distributed on the cell membrane or inside the cell. Based on the number of scanning layers occupied by each protein in Z-stack mode and the step length (0.13 μm), the lengths of these Gag proteins were calculated. The lengths of the Gag proteins distributed on the cell membrane were 2.08, 1.95, 2.34, 1.69, 1.30, and 1.43 μm respectively (standard deviation: 0.40). The lengths of the Gag proteins inside the cell were 1.69, 1.43, 1.30, 1.43, 1.17, and 1.56 μm respectively (standard deviation: 0.18). This part has been cited as supplementary data in the third paragraph of the discussion section (Fig S1).

Fig C. The lengths of the thin, elongated, cord-like Gag proteins distributed on the cell membrane or inside the cell

---

## [Decision Letter · Decision Letter 1]

11 Nov 2025

Dear Dr. Sun,

Thank you for submitting your manuscript to PLOS ONE. After careful consideration, we feel that it has merit but does not fully meet PLOS ONE’s publication criteria as it currently stands. Therefore, we invite you to submit a revised version of the manuscript that addresses the points raised during the review process.

We look forward to receiving your revised manuscript.

Kind regards,

Mauricio Comas-Garcia

Academic Editor

PLOS ONE

**Journal Requirements:**

**Additional Editor Comments:**

All reviewer's have agreed that the new version of the manuscript has been improved. However, I sincerely apologize for not bringing this up sooner, as I should have. The extreme brevity of the Methods and Materials sections makes replication of the experiments impossible and thus inappropriate for PLOS One. For example, this section does not explain how much plasmid was used; it is not clear if the plasmid contains p6 and Env, whether non-tagged Gag was used, the number of replicates, the number of stacks, the length of the movies, or the statistical methods. So I will kindly ask you to improve the details in the experimental section such that your experiments can be replicated. Also, there should be a statistical analysis performed on the data of Fig S1 and given its importance it merits to be moved to the main text. Also, Figures A, B and C in the response to the reviewers are really important and should be in the main text (indicating the proper statistical analysis).

All the best,

Prof. Comas-Garcia

Reviewers' comments:

Reviewer's Responses to Questions

**Comments to the Author**

Reviewer #1: All comments have been addressed

Reviewer #2: All comments have been addressed

2. Is the manuscript technically sound, and do the data support the conclusions?

Reviewer #1: Yes

Reviewer #2: Yes

3. Has the statistical analysis been performed appropriately and rigorously?

Reviewer #1: Yes

Reviewer #2: Yes

4. Have the authors made all data underlying the findings in their manuscript fully available?

Reviewer #1: Yes

Reviewer #2: Yes

5. Is the manuscript presented in an intelligible fashion and written in standard English?

Reviewer #1: Yes

Reviewer #2: Yes

Reviewer #1: The authors replied to all comments and they revised the article according to the advice of the editor and reviewers.

Reviewer #2: All comments have been addressed, and new figure for statistics supports the manuscript.

Only one more minor suggestion: in the discussion part, the authors write: "In this study, the lengths of the thin, elongated, cord-like Gag proteins distributed on the cell membrane or inside the cell are approximately 1-2 µm". This is ambiguous, I think the author means the region of distribution for Gag proteins range is approximately 1-2µm, instead of the protein itself. No matter the monomer Gag alone, or the immature Gag/Gag-pol lattice are far shorter/thinner than 1-2µm (size for a whole HIV-1 virion is usually less than 150nm).

**Do you want your identity to be public for this peer review?** For information about this choice, including consent withdrawal, please see our Privacy Policy

Reviewer #1: No

Reviewer #2: No

---

## [Author Response · Author response to Decision Letter 2]

8 Dec 2025

We appreciate the valuable comments from the editor and reviewers on our manuscript. These comments have offered substantial guidance for our research.

---

## [Editor Report · Decision Letter 2]

10 Dec 2025

Characteristics of spatiotemporal distribution of HIV-1 Gag-containing complexes on the dorsal membrane tracking with live confocal imaging

PONE-D-25-44222R2

Dear Dr. Sun,

We’re pleased to inform you that your manuscript has been judged scientifically suitable for publication and will be formally accepted for publication once it meets all outstanding technical requirements.

Kind regards,

Mauricio Comas-Garcia

Academic Editor

PLOS One
---

## [Editor Report · Acceptance letter]

PONE-D-25-44222R2

PLOS One

Dear Dr. Sun,

I'm pleased to inform you that your manuscript has been deemed suitable for publication in PLOS One. Congratulations! Your manuscript is now being handed over to our production team.

Kind regards,

on behalf of

Dr. Mauricio Comas-Garcia

Academic Editor

PLOS One